# Immunogenicity and Protective Efficacy of Five Vaccines Against Highly Pathogenic Avian Influenza Virus H5N1, Clade 2.3.4.4b, in Fattening Geese

**DOI:** 10.3390/vaccines13040399

**Published:** 2025-04-11

**Authors:** Ronja Piesche, Christophe Cazaban, Leticia Frizzo da Silva, Luis Ramírez-Martínez, Heike Hufen, Martin Beer, Timm Harder, Christian Grund

**Affiliations:** 1Institute of Diagnostic Virology, Friedrich-Loeffler-Institute, 17493 Greifswald, Germany; ronja.piesche@fli.de (R.P.); martin.beer@fli.de (M.B.); 2CEVA Santé Animale, 33500 Libourne, France; christophe.cazaban@ceva.com; 3Zoetis Inc. Veterinary Medicine Research and Development, Kalamazoo, MI 49009, USA; leticia.frizzodasilva@zoetis.com; 4Laboratorio Avi-Mex, Col. de Valle, Ciudad de Mexico 03100, CDMX, Mexico; luis.ramirez@avimex.com.mx; 5Boehringer Ingelheim Vetmedica GmbH, 55218 Ingelheim am Rhein, Germany; heike.hufen@boehringer-ingelheim.com

**Keywords:** vaccination, free range, poultry, protection, HPAIV, H5N1, goose

## Abstract

**Background/Objectives:** The risk of the introduction of highly pathogenic avian influenza virus (HPAIV) in geese breeding and fattening flocks is heightened due to the necessity of free-range access to grazing grounds. This study aimed to evaluate the safety, immunogenicity, and protective efficacy of five commercial vaccines against HPAIV subtype H5N1 (clade 2.3.4.4b) in subadult fattening geese. **Methods:** A prime-boost vaccination trial was conducted using five commercial vaccines, including H5 expressing vaccines of novel technology (subunit, vector, RNA) and whole inactivated virus (WIV) vaccines. Based on serological results, one RNA and one WIV vaccine were selected for a homologous challenge experiment. **Results:** Two vaccines of novel technology (vector, RNA) required a booster dose to raise specific antibodies titers above a threshold of four log_2_ using a hemagglutination inhibition (HI) assay, whereas a subunit vaccine and two WIV vaccines induced seroconversion after primary vaccination. In the challenge experiment, all unvaccinated control geese succumbed to infection by day four. In contrast, all vaccinated geese that had seroconverted exhibited full clinical protection. Although sterile immunity was not achieved, viral excretion was significantly reduced in the vaccinated groups compared to controls. **Conclusions:** Vaccination substantially mitigated the impact of HPAIV H5N1, clade 2.3.4.4b infection in geese, greatly improving animal welfare by preventing severe disease. Additionally, there was a significant reduction in viral burden. Further studies are necessary to verify the potential of these vaccines to reduce susceptibility to infection and virus excretion in order to achieve suppression of the between-flock reproduction number to < 1 in geese flocks at high risk of infection.

## 1. Introduction

The ongoing pandemic of a highly pathogenic avian influenza virus (HPAIV) of subtype H5N1 (goose/Guangdong lineage [gs/GD]), clade 2.3.4.4b, is posing an unprecedented threat, impacting both industrial and small-scale poultry operations [1,2]. Similarly, HPAIV infections in wild birds, particularly those leading to mass mortalities among colony-breeding species, are exerting significant pressure on avian biodiversity [3,4]. The recent increase in spillover infections of HPAIV H5N1 clade 2.3.4.4b to wild mammals and livestock (e.g., dairy cows) further expands the interface of exposure between animals and humans [5]. The increased risk of zoonotic transmission to humans has prompted reconsideration of preventive measures against HPAIV infections in animals.

In Europe, the control of HPAI in poultry and other captive birds has traditionally relied on enhanced biosecurity measures to prevent virus introduction from the environment or wild bird populations [6]. These measures also aim to limit virus spread from infected poultry enterprises and are supported by highly effective diagnostic systems and surveillance strategies for early disease detection [7]. Upon confirmation of infection, immediate restriction measures are enforced, including the culling of affected and suspected flocks, holding closures, restocking bans, and regional trade restrictions. These principles have successfully prevented HPAIV from becoming enzootic in captive birds in Europe for nearly 15 years. However, the current panzootic spread of the clade 2.3.4.4b virus and its established year-round presence in wild birds in Europe have led to EU legislation establishing a legal basis for HPAIV vaccination as an additional preventing measure (EU 2023/361) [8,9].

According to fundamental virological knowledge and experience gained from vaccination campaigns in non-European countries, the success of poultry vaccination programs depends on two critical factors: (i) the antigenic similarity between the vaccine virus strains and the HPAIV strains currently circulating in the field, and (ii) the rigorous monitoring of the vaccinated herds [10,11,12]. Employing vaccines that antigenically match circulating field viruses provides superior efficacy and should limit the risk of emerging virus escape variants [13]. Close-meshed surveillance activities aim at early detection of HPAIV circulating stealthily in vaccinated herds that appear clinically healthy.

In Europe, practical experience with HPAI vaccinations is limited and further insight is needed into the appropriate use of diagnostics that are fit-for-purpose in vaccinated flocks. Vaccination has been conducted across Europe in various poultry species using both laboratory-based approaches (Italy—turkeys, Netherlands—laying hens) and field studies (France—fattening ducks) [14,15,16]. Although geese are a minor poultry species economically, their production can be locally significant. Breeding geese requires sufficient free-range access, increasing the risk of HPAIV introduction. Despite the relatively small number of geese farms, they have been disproportionally affected by HPAIV outbreaks in Europe [11]. Therefore, additional protection through vaccination could have a major impact in geese as a particularly exposed sector of poultry production.

A laboratory-based pilot trial was conducted to assess the immunogenicity and efficacy of selected commercially produced, though not yet licensed, for use in the European Union, vaccines in geese. Additionally, various diagnostic tools were evaluated for the surveillance of vaccinated flocks.

## 2. Materials and Methods

### 2.1. Ethics Statement

All animal experiments were approved by an independent ethics committee of the German Federal State of Mecklenburg-Vorpommerania (LALLF 7221.3-1-037/23-1). All animal trials were conducted in biosafety level 3 (BSL3) animal facilities of the Friedrich-Loeffler-Institute, the Isle of Riems, Greifswald.

### 2.2. Animal Experiments

One-day-old goslings of a commercial breed of German laying geese were obtained directly from a commercial hatchery and housed in stables at Friedrich-Loeffler-Institute (FLI) with continuous access to dry bedding material and bathing facilities. Subsequent testing demonstrated that goslings were seronegative for AIV-specific antibodies and negative in combined oropharyngeal and cloacal swabs using generic M-gene-specific real-time RT-PCR (RT-qPCR). Feed and water were provided ad libitum. Throughout the experiments, the animals were supervised at least twice daily by veterinary staff and animal caretakers. Continuous video supervision and a remote-controlled light monitoring regime were also enabled. Any clinical signs of disease occurring before, during, or after the vaccination, as well as those resulting from the challenge infection in two selected groups, were fully documented. Humane endpoints were defined for significant clinical deterioration (see document Appendix A), including severe neurological deficits and/or loss of vital functions. Animals that reached these endpoints were first anesthetized with a combination of Ketamine-Xylazin (4.4 mg/kg Ketanest^®^ (Pfizer, New York City, NY, USA)) and 2.2 mg/kg Xylavet^®^(cp-Pharma, Burgdorf, Germany) injected into the caudal femoral muscle. Unconscious animals were then euthanized via exsanguination by opening the Sinus occipitalis and subsequent dislocation of the neck.

### 2.3. Vaccines

None of the vaccines employed had been licensed for use in Europe in geese. The vaccines were provided by the producers under Material Transfer Agreement (MTA) and Non-Disclosure Agreements (NDA) for exclusive use at FLI in the trial. The vaccines were kept at FLI under the recommended storage conditions. Vaccines 1-4 expressed a hemagglutinin (HA) H5 of clade 2.3.4.4, while vaccine 5 consisted of baculovirus-expressed H5 with an altered amino acid sequence for broader protection [17].

### 2.4. Experimental Design

Acclimatization periods of six weeks were selected to ascertain that animals were a homogenous group and clinically healthy during the raising period. In addition, this period was included to allow immune maturation of chicks after hatch [18]. Subsequently, animals were randomly separated into six groups of ten birds each (five vaccine groups + one unvaccinated control group) (Table 1). Geese were immunized at six weeks of age with one of the vaccines (Table 1, Figure 1). Due to a delayed disposability, geese of the #BI group were vaccinated in the tenth week of age. For all groups, booster vaccination was administered four weeks after the priming, i.e., at 10 weeks of age for groups #Ceva, #Zoe, #Avi-1 and #Avi-2 and at 14 weeks of age for the #BI group, respectively. The final assessment of immunogenicity and subsequent HPAIV H5N1 challenge was conducted three weeks following booster immunization.

### 2.5. Virus

For challenge experiments, a clade 2.3.4.4b duck pathogenic HPAIV strain (HPAIV H5N1 *A/chicken/Germany-NI/AI 4286/2022*) was used (see Table 2) [19]. The viral stock was derived from the amnion allantoic fluid of inoculated specific pathogen-free (SPF) embryonated chicken eggs. The virus titer was determined in cell culture (immortalized chicken hepatocyte cell line LMH [ATCC: CRL-2117]) as well as in embryonated SPF chicken eggs, both in triplicates and calculated using Reed and Muench’s method [20]. Additional viruses used for serological purposes in this study are listed in Table 2.

### 2.6. Serology

Heparinized blood samples were collected from all geese at indicated timepoints (Figure 1), i.e., before prime immunization, before booster vaccination, 3 weeks after booster vaccination (prior to challenge) and 14 days after the challenge infection at the end of the observation period. Plasma samples were heat inactivated (56 °C, 30 min) and tested using two commercial ELISA test kits for Influenza A-specific antibodies against the NP or H5 protein (ID screen^®^ Influenza A Antibody Competition Multi-species ELISA; ID screen^®^ Influenza competitive subtype-specific H5 kit) according to the manufacturer’s instructions.

Plasma samples were also tested using a hemagglutination inhibition (HI) assay, serum neutralization test (SNT), using antigens listed in Table 2. For HI assays, geese plasma was pre-absorbed with 10% (*v*/*v*) chicken erythrocytes (37 °C, 30 min) and subsequently tested according to standard procedures using chicken red blood cells [21,22]. Following the definition from the standard protocol of the WOAH Terrestrial Manual [22] for diagnostic testing, samples above a threshold of 4 log_2_ were considered HI-positive.

The SNT was performed in a 96-well plate format, producing a serial twofold dilution of the plasma samples. Each 100 µL of plasma dilution was mixed with 100 µL of prediluted virus containing 400 TCID_50_. After incubation for 30 min at 37 °C, a triplicate of 50 µL from each dilution series was transferred to a 96-well plate with a preformed monolayer of the LMH cells. After a 72-h incubation period at 37 °C and 5% CO_2_, the plates were analyzed using light microscopy for viral cytopathic effects.

Arithmetic mean titers and standard deviations (stdv) were calculated from all individuals with positive HI results (>3 [log_2_]) according to the vaccination schedule. Arithmetic means were calculated using Microsoft Excel (Microsoft Corporation, (2018). Microsoft Excel. Retrieved from https://office.microsoft.com/excel, accessed on 17 December 2024) formula (=MITTELWERT(range)), where ‘range’ referred to the selected data set. The standard deviation was calculated using (=STABW.N(range)) for the sample standard deviation, where ‘range’ represents the corresponding data range.

### 2.7. Challenge Experiment

Protection was assessed via challenge experiments using 10 geese per vaccine group. To minimize the number of animals in potentially strenuous challenge trials, two vaccine groups were preselected based on their average serological response measured by HI against the challenge virus after booster immunization (Figure 2). Additionally, an unvaccinated control group of 10 geese was established. A total of 30 geese were inoculated oculo-nasally with 0.5 mL/nostril HPAIV H5N1 A/chicken/Germany-NI/AI 4286/2022 with a titer of 10^6^ TCID_50_/0.5 mL (Table 2) and monitored for up to two weeks, as described in Section 2.2. A clinical scoring system, taking into account the general clinical condition, the appetite and central nervous system disorders, was applied to define the endpoint for each bird (for details, see Appendix A). In addition, the clinical score was used to estimate a clinical index (CI) for each group according to the established Intra venous pathogenicity index (IVPI) for HPAIV [22]), considering animals with mild clinical signs below the endpoint as “sick” [1], animals reaching the endpoint as “severely sick” [2] and birds that succumbed to infection as dead [3]. Each animal on trial had swab samples collected from the oropharynx and cloaca at days 2, 3, 4, 5, 7, 10 and 14 post-challenge. Swabs were transferred into tubes containing 1 mL cell culture medium supplemented with Enrofloxacin (Baytril^®^ (Bayer, Leverkusen, Germany); 20 µg/mL) and stored at −70 °C until further analysis. Tissue samples of brain, lung, duodenum/pancreas, kidney, liver and heart obtained at post-mortems of geese that died within the observation period were collected for virological examination into tubes containing 1 mL of cell culture medium with Enrofloxacin (Baytril^®^; 20 µg/mL) or 10% buffered paraformaldehyde.

### 2.8. Real-Time RT-PCR

RNA was extracted from swab samples and tissues as described elsewhere [19,23] using the Macherey-Nagel NucleoMag^®^ VET-Kit on a Biosprint 96 extraction robot (Qiagen, Hilden, Germany) and eluted in 100 µL elution buffer. Subsequently, RNA extracts were tested by a generic real-time RT-PCR (RT-qPCR) for the presence of influenza A virus matrix gene fragment using the AgPath ID One-Step RT-PCR Kit (Ambion-Applied Biosystems, Austin, TX, USA) [7]. PCR tests were carried out in a CFX96™ Real-Time-System C1000™ thermal cycler (BioRad, Munich, Germany). Estimation of virus genome equivalents (VE) in each sample was based on correlation of the individual Cq values to an intra-assay calibration curve of a defined HPAIV H5N1 virus stock with known TCID_50_ infectivity titers.

### 2.9. Virus Isolation from Clinical Materials

To test for infectivity in swab samples, LMH cells were seeded in 24-well cell culture plates with 10% fetal calf serum (FCS) and 10% Baytril^®^ (Enrofloxacine). Examination of oropharyngeal swabs only was chosen because of the reduced load of bacterial contamination compared to cloacal swabs. Following a 24-h incubation period at 37 °C and 5% CO_2_, 100 µL of each of the 1:10 pre-diluted supernatants of oropharyngeal swabs, filtered through a 0.22 µm filter (Millex^®^ Syringe-driven Filter Unit—0.22 µm) (Sigma-Aldrich, St. Louis, MO, USA), were infected in duplicates onto a confluent monolayer of LMH cells. Microscopic examination of cytopathic effect (CPE), manifested as cell necrosis and plaque formation, was conducted after 72 h incubation at 37 °C and 5% CO_2_. Subsequently, the supernatants from cells showing CPE were examined for virus-induced hemagglutination ability in a hemagglutination assay (HA). A CPE associated with an HA titer >3 log_2_ in the supernatant was considered an indication of residual infectivity. In the case of inconclusive CPE and HA in the first cell passage, a second cell passage on LMH cells was conducted, followed by further HA activity investigation.

### 2.10. Statistics

Differences between antibody titers within the groups and assay methods at various time points were determined using two-tailed Mann–Whitney-U tests with the statistical program GraphPad Prism in version 10.3.1 for Windows, GraphPad Software, Boston, MA, USA, https://www.graphpad.com/. Differences in oropharyngeal virus excretion between the groups subjected to challenge infection were statistically evaluated using an unpaired two-tailed Mann–Whitney-U test and Kruskal–Wallis test, respectively, with the help of the statistical analysis tool SigmaPlot™ version 11 (SigmaPlot Software, Grafiti LLC, Palo Alto, CA, USA). The total excretion of virus per animal was determined by calculating the area under the curve (AUC) for each animal individually and as a sum of all animals per group. Comparisons between vaccinated and control groups were limited to 1–4 days post infection (dpi). The AUC of excretion data was determined using the software package R, version 4.3.2 (RStudio Team. (2023). RStudio: Integrated Development Environment for R. RStudio, PBC). Results were analyzed and visualized using the R packages “pracma”, “dplyr” and “ggplot2”.

## 3. Results

### 3.1. Safety and Immunogenicity

Prior to vaccination, all geese tested seronegative for AI virus-specific antibodies via ELISA, HI and SNT. Combined oropharyngeal/cloacal swab samples were negative for AI virus-specific RNA via RT-qPCR). Vaccination did not induce adverse local or systemic signs in any of the geese.

Seroconversion in vaccinated groups depended on the type of vaccine and diagnostic assay. All geese of the unvaccinated control group remained seronegative in all assays applied (for details of serological tests, see Appendix A).

#### 3.1.1. AI Virus H5-ELISA

The H5 antibody response was assessed using a commercial H5 competition ELISA, where a positive response was defined as having residual reactivity below 50%. Seroconversion after prime immunization was detected only in group #BI, with nine out of ten geese showing a positive response (Figure 2A). Booster immunization further increased H5-ELISA titers in all tested geese within this group. In groups #Zoe and #Avi-2, only two geese in each group tested positive for H5-antibodies following the first vaccine dose. Booster vaccination increased the number of geese with positive H5-antibody response to seven in group #Zoe and eight in group #Avi-2. The H5-antibody response in groups #Ceva and #Avi-1 was undetectable after primary immunization; however, it increased after booster vaccination, with half of the animals (n = 5) of the #Ceva and (n = 4) of the #Avi-1 groups showing a positive response. One and two geese in the #Ceva and #Avi-1 groups, respectively, did not seroconvert even after the booster vaccination.

#### 3.1.2. AI Virus NP-ELISA

Geese immunized with novel technology vaccines from groups #Ceva, #Avi-2, and #BI, which lacked AI virus nucleoprotein (NP) and other AI virus proteins except HA, did not develop detectable NP-specific antibodies after the primary and booster vaccinations. In contrast, all geese vaccinated with whole virus antigens (groups #Zoe and #Avi-1) showed NP-specific reactivity already after the primary vaccination (Figure 2B).

#### 3.1.3. HI Assay

Plasma samples were tested via an HI assay using an antigen homologous to the challenge virus HPAIV H5N1 A/chicken/Germany-NI/AI 4286/2022. Seroconversion after prime immunization was detected in groups #Zoe (n = 10) and #BI (n = 5), with HI arithmetic mean titers of 5.1 ± 0.7 (stdv) [log_2_] and 4.3 ± 1.4 [log_2_] (for details, see Appendix A). Booster vaccination further increased HI titers to 8.1 ± 2.8 [log_2_] in the #Ceva and 7.0 ± 1.5 [log_2_] in the #Zoe group, developing the highest mean HI-titers. Primary vaccination induced antibody titers exceeding HI titers of three [log_2_] in three out of ten geese in both groups #Ceva and #Avi-1 (Figure 2C; Appendix A). Booster vaccination increased the number of geese with a positive H5-antibody response to seven in group #Ceva (8.1 ± 2.8 [log_2_]) and ten in group #Zoe (7.0 ± 1.5 [log_2_]). However, two animals from the #Ceva group remained seronegative with HI. Vaccines #Avi-1 and #BI induced HI arithmetic mean titers of 5.3 ± 1.7 (stdv) [log_2_] (see Appendix A for statistical data) and 4.9 ± 1.2 [log_2_], respectively. No HI titers were detected in the group #Avi-2 after either prime or booster vaccination. Antibody cross-reactivity was further investigated using another H5N1 clade 2.3.4.4b antigen from 2021, as well as an antigen from an H5N1 virus clade 1.0 heterologous to the challenge virus (see Table 2), Appendix A). This analysis only included groups that reacted positively (>3 [log_2_]) to the homologous HI assay antigen, thereby excluding #Avi-2. This reactivity was notable already after prime immunization in individual birds and only in geese that were reactive with the clade 2.3.4.4 antigen (Table 2) (see Appendix A). The reactivity to the clade 1.0 virus was most abundant for the #BI vaccinated group (n = 10) and increased after booster immunization from 6.8 ± 1.3 [log_2_] to 8.1 ± 1.5 [log_2_], clearly exceeding titers against clade 2.3.4.4b antigens from 2022 and 2021, (deviation from other groups tested: *p* < 0.0001–0.001). In the #Ceva group, the number of geese reactive with the clade 1.0 antigen increased from three after prime to six after booster vaccination. In group #Zoe, all ten geese initially reacted but showed no booster-induced increase in antibody titers. In group #Avi-1, only three geese had an initial reactivity, with no booster-associated increase. In these latter three groups, reactivity was superior against clade 2.3.4.4b antigens (Figure 3; Appendix A).

HI-reactivity was also tested by employing an NDV antigen. All vaccines that contained NDV components, i.e., group #Avi-2 and #BI, induced NDV-specific antibodies already after prime immunization (HI-titer: #Avi-2 8.3 ± 1.7 [log_2_]; #BI 8.2 ± 2.7 [log_2_]). These titers increased after booster vaccinations (HI-titer: #Avi-2 9.9 ± 1.6 [log_2_]; #BI 9.3 ± 1.6 [log_2_]) (deviation to other groups tested: *p* < 0.0001) (Figure 2D; Appendix A). In contrast, #Ceva, #Zoe, and Avi-1 remained HI-negative (<2 [log_2_]) in NDV HI, since they did not contain this antigen (Figure 2D and Figure 3; Appendix A).

#### 3.1.4. SNT Assay

All groups were tested for their seroneutralization activity against the 2.3.4.4b antigen (Figure 2E, Appendix A), homologous to the challenge virus. The results corroborate those of the HI assay. Vaccines #Zoe (SNT arithmetic mean 3.4 ± 1.2 [log_2_], #Avi-1 and #BI (both 2.3 ± 0.6 [log_2_])) induced measurable SNT-titers after first dose vaccination. In contrast, no SNT reactivity was observed after priming with the #Ceva vaccine (deviation #Ceva to other groups tested: *p* < 0.0001). Booster immunization induced an increase of SNT-titers across all tested groups except #Avi-2: After the booster, the highest SNT-titers were observed in groups #Ceva (6.6 ± 2.3 [log_2_]) and #Zoe (6± 0.8 [log_2_]), followed by groups #Avi-1 and #BI with 4.1 ± 1.2 [log_2_] and 3.8 ± 0.9 [log_2_] (deviation to #Zoe: *p* = 0.0002).

HI and SNT assays detect antibodies that interact with native viral components, therefore, they have biologically relevant readouts. Based on these results, vaccine candidates #Ceva and #Zoe were selected for a subsequent challenge experiment.

### 3.2. Protective Efficacy Against Homologous Challenge

#### 3.2.1. Full Clinical Protection in Seropositive Vaccinees

Geese in the unvaccinated control group started to show clinical signs three days post infection (dpi) onwards. Initially, one animal exhibited a reduced general condition, and a few others had light green, watery feces. By four dpi, the health of all animals had deteriorated, with four geese in very poor condition. An additional five geese presented severe central nervous clinical signs, including torticollis and loss of flight reactions. These nine birds were euthanized on four dpi, including the last goose, which, although mildly affected at that timepoint, was expected to die during the night and was euthanized to prevent suffering from isolation. For clinical index (CI) estimation, this last animal was considered dead the following day, resulting in a CI of 2.13 for the control group. In vaccine group #Ceva, 9 out of 10 challenged birds showed no signs of disease during the 14-day observation period (CI = 0.24). However, the one goose that remained seronegative in all tests, including the SNT, died peracutely on four dpi. In the challenged group #Zoe, none of the 10 test animals showed any clinical reactions (CI = 0).

#### 3.2.2. Vaccinations Reduced Virus Shedding

Despite the initial absence of clinical signs, viral shedding was detected via RT-qPCR in swabs of all unvaccinated geese at two dpi (Figure 4C). The virus load in the oropharyngeal swabs ranged from 3.75 × 10^5^ to 6.23 × 10^7^ (VE/mL). The virus load increased to 3.61 to 5 × 10^8^ (VE/mL) in individual animals until three dpi and remained high on four dpi, when animals were euthanized (Figure 4, Appendix A). A similar excretion pattern was observed in cloacal swabs but with generally lower viral loads, averaging between 4.31 × 10^3^ to 4.11 × 10^6^ (VE/mL) (Figure 4, Appendix A). In vaccine group #Ceva, oropharyngeal swab samples of seven individual animals were positive on two dpi with virus loads between 4.93 × 10^3^ and 1.07 × 10^4^ (VE/mL). By three and four dpi, all nine clinically healthy geese shed the virus, with one bird reaching a viral load as high as 1.94 × 10^7^ (VE/mL) on three dpi. The single seronegative goose in this group, which succumbed to infection on four dpi, shed the virus to levels comparable to the unvaccinated control group from 1.69 × 10^7^ to 2.39 × 10^8^ VE/mL detected in oropharyngeal swab samples as early as two dpi. In the remaining clinically healthy geese, virus excretion gradually subsided until 10 dpi, when all nine surviving birds tested positive for the last time with very low viral loads (Figure 4, Appendix A). Cloacal swabs of this group showed excretion kinetics similar to the oropharyngeal results, with a gradual increase in the number of positive birds and viral load until 4 dpi, returning to negative PCR results by 14 dpi. Shedding patterns in the #Zoe vaccine group were similar to those in the #Ceva vaccine group, with virus shedding observed between 2 and 10 dpi. Statistical comparison of excretion patterns, analyzed by estimating the area under the curve (AUC), revealed that both vaccines significantly reduced excretion (#Ceva *p* = 0.0003; #Zoe *p* < 0.0001) compared to unvaccinated controls, excluding the serological non-responder in group #Ceva (see Appendix A, time period one to four dpi). No significant differences were found between the two vaccine groups when comparing the time period until 4 dpi or until 14 dpi (Figure 4A, Appendix A).

#### 3.2.3. Failure to Isolate Virus in Cell Culture from Swabs of Vaccinated Geese

In addition to RT-qPCR, positive oropharyngeal swab samples were analyzed for infectivity via cultivation on LMH cells. An infectious virus was recovered from numerous swab samples from unvaccinated control geese between two and four dpi (8/10, 5/10 and 3/10, respectively) (Figure 4B and Appendix A). In contrast, no infectious virus was recovered from any swabs obtained from clinically healthy vaccinated geese, except the seronegative goose in the #Ceva group that died on four dpi, where the infectious virus was recovered from swab samples from two and three dpi (Figure 4B, Appendix A).

#### 3.2.4. AIV Infection Was Cleared in Vaccinated, Seropositive Geese

At the end of the observation period, organ samples of euthanized geese were analyzed. This included geese that succumbed to infection by 4 dpi and vaccinated geese that survived challenge infection without clinical signs until 14 dpi. The results confirmed systemic infection, with the highest viral load detected in the brain tissues of clinically diseased geese (3.26 × 10^8^ to 2.79 × 10^10^ VE/mL), regardless of whether they showed clinical CNS disorders (see Appendix A). Interestingly, fat tissues from individual animals also yielded high viral RNA loads (1.12 × 10^6^ to 7.02 × 10^7^ VE/mL). In contrast, none of the organ samples from the vaccinated animals sacrificed on 14 dpi yielded any PCR-positive results in the organs, indicating full clearance of the infection.

#### 3.2.5. NP-Specific Seroconversion as a Serological DIVA Surveillance Tool

Animals receiving marker vaccines that lacked viral NP (groups #Ceva, #Avi-2, #BI) did not seroconvert following primary and booster immunizations in an NP-specific ELISA, while all geese receiving whole virus vaccines seroconverted after primary vaccination (see Figure 2B, Appendix A). Post-challenge plasma samples taken on 14 dpi revealed that all nine geese in the #Ceva vaccine group seroconverted in the NP-ELISA, whereas reactivity in the #Zoe vaccine group, which already tested positive before challenge, remained stable. Antibody reactivity increased in both vaccinated groups as measured via H5 ELISA and via HI with the antigen homologous to the challenge virus (Figure 5).

## 4. Discussion

In our study, five vaccine candidates were selected for testing based on (i) their inclusion in the EFSA’s list of available potential vaccines [11] and (ii) their potential future relevance for the European market, with some of them being produced in Europe. Furthermore, some of these vaccines are already undergoing assessment or are already being used in a few European countries. The latter includes the #Ceva and #BI vaccines, which are employed in vaccination campaigns of fattening ducks in France [14,15,16]. Our results indicate that the primary antibody response to five commercial AI virus H5 vaccines in subadult fattening geese was weak after one vaccine dose administration. However, this response was significantly boosted by a second application of the same vaccine.

To assess vaccine immunogenicity, a comparison of four serological assays with different test principles proved valuable. The competitive subtype-specific H5 ELISA was an effective, non-species-specific tool for detecting vaccine-induced antibodies. Its rapid test format made it suitable for tracking antibody formation at various stages of the vaccination process. However, the biological relevance of the target epitope in the competitive H5 ELISA is not defined. Therefore, complementary diagnostic methods were used; the HI assay, which indicates antibody-mediated interference of virus-receptor binding, is considered a better surrogate marker for protection, particularly when the homologous antigen is used [12]. A possible discrepancy between H5-ELISA reactivity and HI becomes evident when comparing the data in the #BI group. Using H5-ELISA, the #BI-vaccine was considered the most immunogenetic (Figure 2A, Appendix A) (deviation to all other groups: *p* < 0.0001–0.01). However, using the HI assay with an antigen homologous to the challenge virus, reactivity was considerably lower than in group #Ceva and significantly lower than in group #Zoe (*p* = 0.002). Similarly, the #Avi-2 group showed the second-best reactivity in H5-ELISA, but plasma from this group failed to show any HI-reactivity with the AI virus H5 clade 2.3.4.4b antigen. This might in part to an antigenic mismatch of the HA-antigen in the vaccine and in the test procedure. Using the clade 1.0 antigen revealed that the #BI vaccine induced a stronger reactivity toward this historical antigen. This broad cross-reactivity induced by the #BI vaccine is remarkable and might be beneficial for broad application against H5 clades of the gs/GD lineage. For intervention to the currently circulating HPAIV H5 clade 2.3.4.4b, a clade-specific, tailored antigen seems more efficient. This conclusion is further supported by the SNT, which corroborated the HI results with the clade 2.3.4.4b antigen.

Our results also highlight the limitations of using the HI as a prognostic marker for the protection status of a bird, as previously attempted [24]. Although the HI assay provided valuable insights into strain/clade-specific immune responses when different antigens were used (Table 2), it is important to recognize that antibody levels alone do not necessarily correlate with clinical protective efficacy. In our study, at least 3 out of 10 animals of group #Ceva showed homologous HI titers of log_2_ < four, which are often interpreted as insufficient protection. However, only one of these geese ultimately proved not to be protected after challenge infection. This is similar to what was observed with the #AVI-2 vaccine, which, despite the absence of HI titers, protected chickens against challenge with the HPAI virus A/turkey/Italy/21VIR 9520-3/2021 strain, belonging to clade 2.3.4.4b [Ramirez-Martínez, unpublished 2025]. Therefore, HI results can be used to estimate the overall protective efficacy of vaccines in a flock and to facilitate comparisons between them. At the individual level, however, the correlation of HI in the lower titer ranges with clinical protection remains weak. In particular, this might be true for the #Ceva vaccine, as this vaccine is based on a new RNA technology, expressing the antigen intracellularly, thereby enabling MHC-1 presentation and the induction of cellular, i.e., CD8+, immune response [25,26,27,28]. The methods described here, in particular the HI assay, can be used to describe the humoral immune response triggered by vaccines. However, based on our analyses and that of a previous study [13], it has become clear that no general threshold of HI-titers indicates protection. In particular, at the individual level, correlating low HI with clinical protection remains difficult, as Spackman et al. [24] and others [29] have already shown. Nevertheless, the observation that a serologically unreactive goose succumbed to infection like the unvaccinated geese might indicate that the level of antibodies is not the key for clinical protection but rather the priming of the immune system, enabling a rapid response upon recurrent contact/infection [18]. Further in vitro studies elucidating the induced memory effect of such vaccines might help to elucidate such a mechanism.

The lack of sufficient quantification methods for avian cellular immune responses applicable to routine diagnostics, apart from IFN-α staining [30,31], impedes developing alternative immunological marker systems as correlates of protection [12]. Meanwhile, even though technically intricate and labor-intensive, the HI assay can be considered the method of choice to test the antigenic match of a vaccine. Plasma sample processing with pre-incubation of plasma with erythrocytes proved to reduce the non-specific reaction often observed with serum from waterfowl species. In this context, testing sera in parallel for NDV antibodies was a good indication of the specificity of the processed plasma samples, as three vaccines did not contain NDV components and remained seronegative. For vaccines #Avi-2 and #BI, on the other hand, it was another indicator of proper vaccine administration. Altogether, considering the practical limitations of the HI assays, the H5-ELISA appears more appropriate for large-scale surveillance once the correlation of immunogenicity of certain vaccines has been established.

The NP-ELISA, as the fourth serological assay, was applied to verify the concept of Differentiating Infected from Vaccinated Animals (DIVA). None of the geese immunized with any of the three vaccines that lacked the AIV-NP protein (#Ceva, # AVI2, #BI) were reactive within the NP-ELISA. Furthermore, following challenge infection, all geese immunized with the #Ceva vaccine seroconverted using the NP-ELISA. This finding aligns with shedding data, demonstrating productive infection in vaccinated geese after challenge, and corroborates the concept of the NP-ELISA as a DIVA tool for retrospective surveillance.

For risk-based surveillance, however, direct detection of the virus is mandatory to differentiate notifiable AI virus H5/H7 infections from infections with other subtypes like AI virus H6 or H9, which regularly circulate in poultry. Using the generic RT-qPCR, targeting the AI virus matrix (M)-gene, viral RNA was detected in swab samples collected from the oropharynx and to a lesser extent from cloacal swabs (Appendix A), identifying infected geese as early as two dpi, i.e., two days before the onset of the disease in unvaccinated control geese. The test provides evidence that vaccinated geese of both groups shed virus between 2 and 10 dpi. Yet, virus shedding was significantly reduced compared to naive geese (Figure 4). The detected residual virus shedding in clinically healthy birds confirms previous results in other poultry species and highlights a key challenge of AI virus vaccination regimes, which might facilitate silent virus spread within and between flocks [32,33]. Identifying field virus-infected individual birds that might act as effective transmitters and spreaders of the virus within a flock is considered of utmost importance for any surveillance strategy of vaccinated flocks [34]. Concerning the efficacy of the vaccine, a key factor is the decrease in susceptibility and transmissibility affecting virus spread within and between farms. In this respect, our results might hold some promising perspectives: Even though RT-qPCR results point to viral RNA shedding in vaccinated geese, no infectious virus was recovered in oropharyngeal swabs using an LMH cell culture from infected vaccinated geese, while virus isolation was successful from swabs of unvaccinated control animals (Figure 4B, Appendix A). This might be in part due to virus neutralization by mucosal antibodies [35]. Otherwise, this might reflect an insufficient detection limit of the LMH cell culture system used here. Ultimately, further studies focusing on the basic reproduction number, R_0_, are needed to investigate whether this reduced virus shedding has an impact on virus transmission between geese. This index indicates the average number of secondary cases caused by one individual throughout its infectious phase [36,37]. An effective vaccine should aim at reducing transmission to a level of R_0_ < 1. Based on estimates of R_0_, the anticipated protection efficacy of candidate vaccines can be compared to develop more effective disease control strategies. The data obtained here will be used to select candidates for further transmission experiments. Moreover, the protective efficacy of the vaccines that were not challenged remains to be determined. Future studies should include challenge trials for these vaccines that demonstrated immunogenicity to achieve a more complete evaluation of their potential role in disease control.

## 5. Conclusions

In conclusion, four out of five tested commercial AIV H5 vaccines were immunogenic in geese after booster immunization, and two candidate vaccines induced the highest antibody titers via HI and SNT-protected animals against clinical disease. Although vaccinated geese remained susceptible to HPAIV H5 challenge infection, virus excretion was significantly reduced, and no infectivity was recoverable in a cell culture-based system from oropharyngeal swab samples of these birds. Our study provides valuable baseline data for evaluating vaccines against HPAIV in geese in Europe. Full clinical protection is a readily achievable goal following a prime-boost vaccination scheme. However, duration of immunity was not investigated here, but re-vaccination should likely be considered for geese that are kept for longer than four months. Sterile immunity was not induced, and the effects of vaccination on suppressing transmission are currently under investigation. Studies on the reduction of virus transmissibility to and between vaccinated geese by estimating R_0_ are recommended for a full characterization of the epidemiological efficacy of such vaccines. In any case, close-meshed surveillance of vaccinated geese flocks will be mandatory; in this respect, the inclusion of environmental sample matrices, in particular bathing water, into sampling schemes is advisable.

## Figures and Tables

**Figure 1 vaccines-13-00399-f001:**
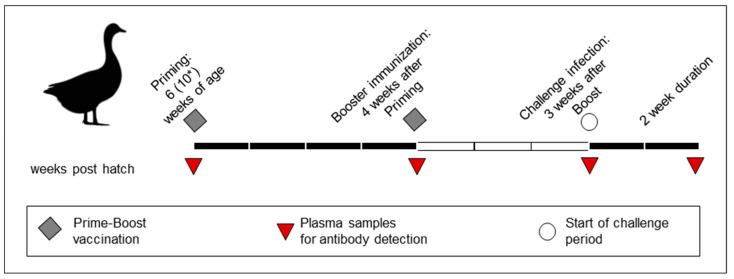
Vaccination scheme. Vaccination included a prime immunization followed by a second booster immunization 4 weeks later. While four of the groups received the first immunization at 6 weeks of age, due to a delay during international shipment, the #BI group was first vaccinated at 10 weeks of age (*). A sixth group was kept under the same housing conditions but did not receive any vaccination (unvaccinated control group). Plasma samples taken at each vaccination time point and three weeks after the booster vaccination were used to determine the antibody response. Two vaccine groups with the highest HI antibody titers and the unvaccinated control group were exposed to a challenge infection with an observation period of up to 2 weeks, depending on the clinical status.

**Figure 2 vaccines-13-00399-f002:**
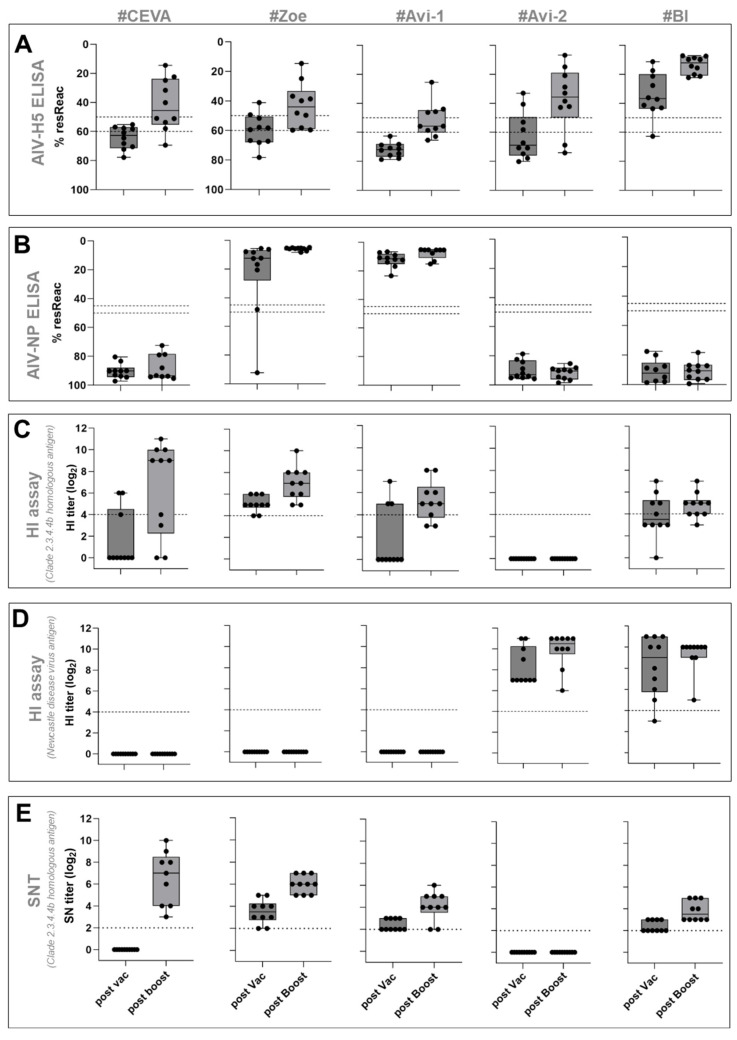
Comparative analysis of the antibody response after prime and booster vaccination with one of five commercial vaccines in five different serological assays. Plasma samples were taken after prime vaccination and three weeks after booster immunization and tested for the presence of AI virus antibodies through H5-ELISA (**A**), NP-ELISA (**B**), HI against H5 antigen (**C**), HI against Newcastle virus antigen (**D**) and SNT (**E**). Dotted lines on graphs (**A**–**E**) mark the respective cut-off, separating negative (below dotted line) from positive titers (above dotted line) (“indeterminate” area between dotted lines for ELISAs). Box-plots showing 25–75th percentiles and median titers.

**Figure 3 vaccines-13-00399-f003:**
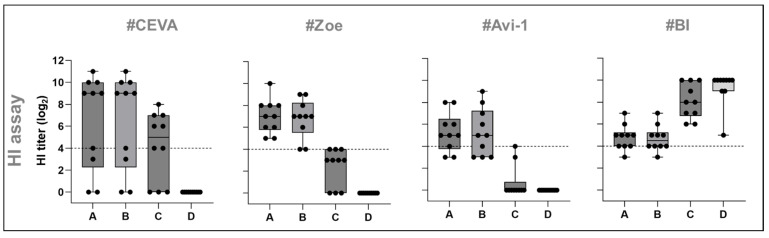
Overview of plasma reactivity in HI assays against different antigens. Plasma samples from four vaccine groups were tested after booster immunization via HI assay against antigens of clade 2.3.4.4b: (A) A/chicken/Germany-NI/AI 4286/2022 and (B) A/chicken/Germany-SH/AI08298/2021. Heterologous antigen originated from clade 1.0 (C) A/chicken/Vietnam/P41-05 (R75/05). In addition, an NDV antigen was used: (D) Genotype 2.II (NDV Clone 30). Dotted lines on graphs mark the respective cut-off (4 log_2_) [22], separating negative (below dotted line) from positive titers (above dotted line).

**Figure 4 vaccines-13-00399-f004:**
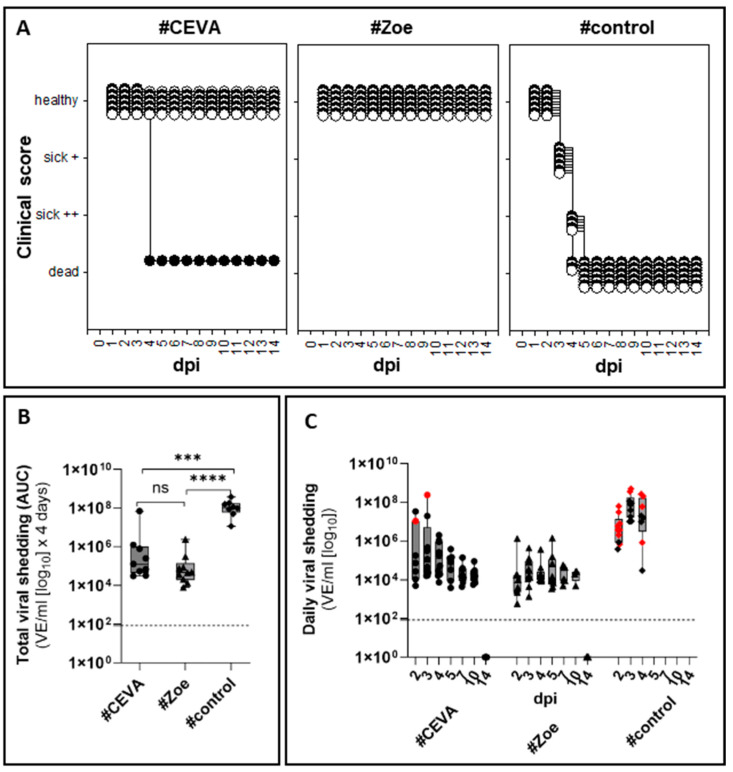
Comparison of Clinical scores (**A**) total (**B**) and individual daily (**C**) oropharyngeal viral shedding of infected groups. Individual clinical status was scored daily between 1 and 14 dpi (**A**). Total oropharyngeal shedding per goose (VE/mL [log_10_]) was compared up to four days post infection, measured by calculating the area under the curve (AUC) values of all animals of the three challenged groups (**B**). The viral load in VE/mL [log_10_] per oropharyngeal swab of each individual animal per day was plotted in (**C**). Colored symbols (red) indicate that infectious virus was recovered from these samples in LMH cell cultures, from swabs derived from the control group only as well as from one non-responder of the group. The levels of significance are indicated by *** (*p* < 0.001), **** (*p* < 0.0001), and ns (not significant).

**Figure 5 vaccines-13-00399-f005:**
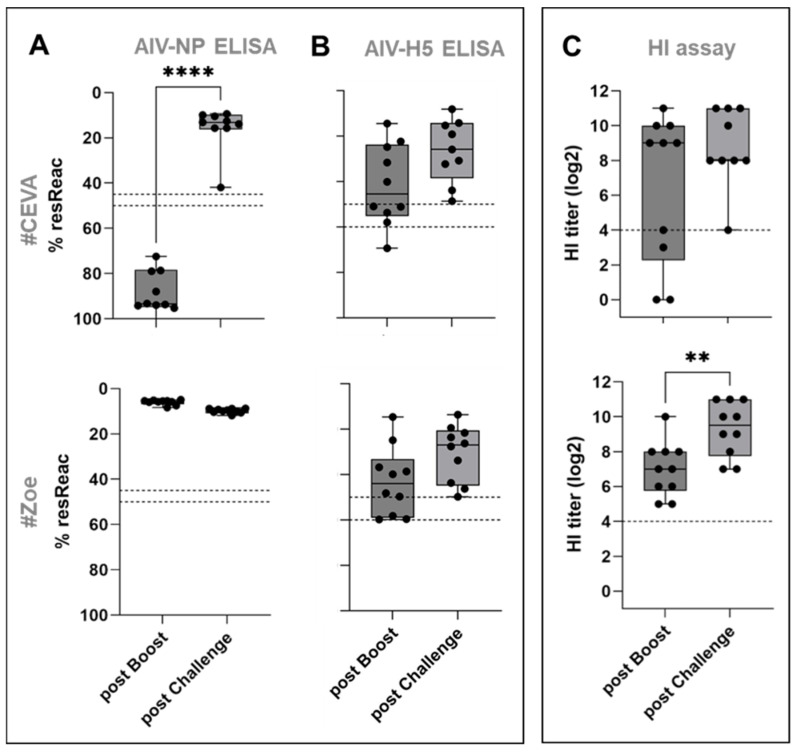
Antibody responses in vaccinated geese following HPAIV H5N1 challenge infection. Sero-reactivity was tested using NP-(**A**) and H5-specific ELISA (**B**) as well as HI (**C**). Significance levels are indicated by ** (*p* < 0.01) and **** (*p* < 0.0001).

**Table 1 vaccines-13-00399-t001:** Characteristics of vaccines used.

Group	Vaccine Producer	Vaccine Name	Batch Number	Type (gs/GD Clade)	Dose and Application	
#Ceva	CEVA	RESPONS AI H5	Lot 0409LF	Amplicon (H5) [2.3.4.4b]	0.2 mL	i.m. caudal femoral muscle
#Zoe	Zoetis		NE 69,521 LO523AS03	WIV (H5N2) [2.3.4.4b]	0.5 ml	s.c. neck fold
#Avi-1	Avimex	Vaxigen Flu H5N8 clado 2.3.4.4	Reg B-0258-131	WIV (H5N8) [2.3.4.4b]	0.5 mL	s.c. neck fold
#Avi-2	Avimex	KNewH5	Lote E.PM-2304	recNDV (H5) [2.3.4.4b]	0.5 mL	s.c. neck fold
#BI	Boehringer	Volvac B.E.S.T. AI + ND	2307011A	Baculovirus-based expression system of an inserted optimized H5 sequence for antigen formulation	0.5 mL	s.c. neck fold

**Table 2 vaccines-13-00399-t002:** List of viruses used in challenge trials and in serology.

**Antigen**	**Sub-and Phenotype**	**Clade**	**Isolate**	**Sequence Accession**
A	HP H5N1	2.3.4.4b	A/chicken/Germany-NI/AI 4286/2022	Epi16096050
B	HP H5N1	2.3.4.4b	A/chicken/Germany-SH/AI08298/2021	Epi18006820
C	HP H5N1	1.0	A/chicken/Vietnam/P41-05/2005 (R75/05)	Epi13970
D	NDV	Genotype 2.II	NDV LaSota	ON713864

## Data Availability

All curated data are included in the Appendix A accompanying this manuscript. Additional data are available upon request from the corresponding authors.

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
