# Peer review of "Immunogenicity and Protective Efficacy of Five Vaccines Against Highly Pathogenic Avian Influenza Virus H5N1, Clade 2.3.4.4b, in Fattening Geese"

_vaccines, 2025, doi:10.3390/vaccines13040399_

Round 1
Reviewer 1 Report
Comments and Suggestions for Authors
This article by Piesche et al. evaluated immunogenicity of five different HPAI H5N1 vaccines in geese. Overall, this article is accomplished and written nicely. There are certain comments to improve its quality and clarity:
- Title is incomplete: immunogenicity and protective efficacy of what? Revise this.
- In abstract, 4 log2 is mentioned as the threshold. Not sure what is ‘threshold’ and how it was defined.
- In introduction, it was mentioned that these vaccines are commercial but not licensed. Clarify.
- In methods, humane endpoint is subjective. Please specify what does it mean by significant clinical deterioration? What kind of loss of vital functions and what do the researchers obtained to make a decision of humane endpoint?
- What did mock vaccine include? Mock vaccinated (means PBS or something else) or nonvaccinated? Confirm and keep consistent throughout the manuscript.
- Revise figure 1 to accommodate the delayed vaccine and challenge in BI group. What is the effect of age at first vaccination? That needs to be discussed.
- Why the BI vaccine had HI titer against NDV antigen?
- In results, provide the % of animals that seroconverted to give a better perspective.
- What is the limit of detection for SNT, including it in figure?
- Group-specific statistics (e.g., deviation for Zoe or CEVA) are not clear. Show those data with statistical analysis in supplementary.
- The clinical scores and group averages are not clear. Define the scoring criteria and scores in more detail with a supplementary table.
- Figure 4A, is it data of 4 days? In Figure 4B, what does the red color indicate? The total vs individual shedding is not clear.
- Discuss, what is likely to be providing protection in CEVA vaccine then? More discussion or hypothesis are needed for the protection in birds that lack HI titers.
- As there are co-authors from the all companies who produced the vaccine, how come there is no conflict of interest in the paper?
Reviewer 2 Report
Comments and Suggestions for Authors
Comments:
The manuscript by Piesche et al. evaluated the safety, immunogenicity, and protective efficacy of five commercial vaccines against the HPAIV subtype H5N1 in fattening geese. This manuscript offered promising data supporting the potential of vaccination for HPAIV control in geese, specifically in Europe.
Specific comments:
- Why did you choose these five vaccines? What are selection criteria? Please discuss more in the manuscript.
- Non-vaccinated control geese data in Figure 2 is missing.
- Why samples for serological assays in #BI group were collected at week 14 and 17, while others were collected at 10 and 13? It is challenging to directly compare results across all groups because of the confounding factor introduced by the #BI group's delayed sample collection.
- The presentation of protective efficacy results could be improved for clarity and ease of interpretation. More concise tables (clinical score) and figures (survival curve), possibly with statistical significance indicated directly on the figures would improve readability.
- Based on all the data you got in this study, what helpful information could be provided when vaccines against HPAIV were approved to use in Europe? Please add it in the conclusion section.
- Format error found in lines 396-398.
Round 2
Reviewer 1 Report
Comments and Suggestions for Authors
All the comments raised earlier are addressed.
Reviewer 2 Report
Comments and Suggestions for Authors
No further comments.